# COVID-19 in Children with Down Syndrome: Data from the Trisomy 21 Research Society Survey

**DOI:** 10.3390/jcm10215125

**Published:** 2021-10-31

**Authors:** David Emes, Anke Hüls, Nicole Baumer, Mara Dierssen, Shiela Puri, Lauren Russell, Stephanie L. Sherman, Andre Strydom, Stefania Bargagna, Ana Cláudia Brandão, Alberto C. S. Costa, Patrick T. Feany, Brian Allen Chicoine, Sujay Ghosh, Anne-Sophie Rebillat, Giuseppina Sgandurra, Diletta Valentini, Tilman R. Rohrer, Johannes Levin, Monica Lakhanpaul

**Affiliations:** 1Department of Infectious Disease Epidemiology, Faculty of Epidemiology and Public Health, London School of Hygiene and Tropical Medicine, London WC1E 7HT, UK; david.emes@lshtm.ac.uk; 2Department of Epidemiology, Rollins School of Public Health, Emory University, Atlanta, GA 30322, USA; anke.huels@emory.edu (A.H.); lauren.russell2@emory.edu (L.R.); patrick.feany@emory.edu (P.T.F.); 3Gangarosa Department of Environmental Health, Rollins School of Public Health, Emory University, Atlanta, GA 30322, USA; 4Boston Children’s Hospital, Harvard Medical School, Boston, MA 02115, USA; Nicole.Baumer@childrens.harvard.edu; 5Centre for Genomic Regulation (CRG), The Barcelona Institute of Science and Technology, 08003 Barcelona, Spain; mara.dierssen@crg.eu; 6Universitat Pompeu Fabra (UPF), 08002 Barcelona, Spain; 7Centro de Investigación Biomédica en Red de Enfermedades Raras (CIBERER), 28029 Madrid, Spain; 8Down Syndrome Medical Interest Group UK, Leeds Community Healthcare NHS Trust, Teddington TW11 9PS, UK; s.puri@nhs.net; 9Department of Human Genetics, School of Medicine, Emory University, Atlanta, GA 30322, USA; ssherma@emory.edu; 10Department of Forensic and Neurodevelopmental Sciences, Institute of Psychiatry, Psychology, and Neuroscience, King’s College London, London WC2R 2LS, UK; andre.strydom@kcl.ac.uk; 11The London Down Syndrome (LonDownS) Consortium, London WC2R 2LS, UK; 12South London and the Maudsley NHS Foundation Trust, London WC2R 2LS, UK; 13Department of Developmental Neuroscience, IRCCS Fondazione Stella Maris, 56128 Pisa, Italy; sbargagna@fsm.unipi.it (S.B.); g.sgandurra@fsm.unipi.it (G.S.); 14Hospital Israelita Albert Einstein, ACB, Sao Paulo 05652-900, SP, Brazil; trescodemes@gmail.com; 15Departments of Pediatrics and of Psychiatry, School of Medicine, Case Western Reserve University, ACSC, Cleveland, OH 44106, USA; alberto.costa@case.edu; 16Advocate Medical Group, Adult Down Syndrome Center, Park Ridge, IL 60068, USA; brian.chicoine@aah.org; 17Cytogenetics and Genomics Research Unit, Department of Zoology, University of Calcutta, Kolkata 700 073, West Bengal, India; sgzoo@caluniv.ac.in; 18Institut Jérôme Lejeune, 75015 Paris, France; annesophie.rebillat@institutlejeune.org; 19Department of Clinical and Experimental Medicine, University of Pisa, 56126 Pisa, Italy; 20Pediatric Unit, Bambino Gesù Children’s Hospital, IRCCS, 00146 Rome, Italy; diletta.valentini@opbg.net; 21Division of Pediatric Endocrinology, Saarland University Medical Center, 66421 Homburg/Saar, Germany; Tilman.Rohrer@uks.eu; 22Department of Neurology, Ludwig-Maximilians-Universität München, 80539 Munich, Germany; johannes.levin@lmu.edu.de; 23German Center for Neurodegenerative Diseases, Site Munich, 81377 Munich, Germany; 24Munich Cluster for Systems Neurology (SyNergy), 81377 Munich, Germany; 25Population, Policy and Practice Department, Great Ormond Street Institute of Child Health, UCL, London WC1N 1EH, UK; 26Whittington NHS Trust, London N19 5NF, UK; 27Down Syndrome Medical Interest Group, Teddington TW11 9PS, UK

**Keywords:** COVID-19, Down Syndrome, paediatrics

## Abstract

Adults with Down Syndrome (DS) are at higher risk for severe outcomes of coronavirus disease 2019 (COVID-19) than the general population, but evidence is required to understand the risks for children with DS, which is necessary to inform COVID-19 shielding advice and vaccination priorities. We aimed to determine the epidemiological and clinical characteristics of COVID-19 in children with DS. Using data from an international survey obtained from a range of countries and control data from the United States, we compared the prevalence of symptoms and medical complications and risk factors for severe outcomes between DS and non-DS paediatric populations with COVID-19. Hospitalised COVID-19 patients <18 years with DS had a higher incidence of respiratory symptoms, fever, and several medical complications from COVID-19 than control patients without DS <18 years. Older age, obesity, and epilepsy were significant risk factors for hospitalisation among paediatric COVID-19 patients with DS, and age and thyroid disorder were significant risk factors for acute respiratory distress syndrome. Mortality rates were low in all paediatric COVID-19 patients (with and without DS), contrasting with previous findings in adults with DS (who exhibit higher mortality than those without DS). Children with DS are at increased risk for more severe presentations of COVID-19. Efforts should be made to ensure the comprehensive and early detection of COVID-19 in this population and to identify children with DS who present comorbidities that pose a risk for a severe course of COVID-19. Our results emphasize the importance of vaccinating children with DS as soon as they become eligible.

## 1. Introduction

Down Syndrome (DS) is associated with genetic factors and medical comorbidities that may lead to increased vulnerability to a severe course of coronavirus disease 2019 (COVID-19)—the illness caused by the severe acute respiratory syndrome coronavirus 2 (SARS-CoV-2). Immune dysregulation among individuals with DS increases vulnerability to viral infections, while anatomical airway features increase the vulnerability to respiratory illnesses [1,2,3]. Respiratory illnesses are a major cause of premature mortality in people with DS [4,5], and children with DS show a high susceptibility to recurrent respiratory infections [6]. Indeed, many non-respiratory comorbidities of DS, such as obesity, diabetes, or heart conditions, are known risk factors for morbidity and mortality from COVID-19 in the general population [2,3].

Unlike in the adult population, limited data are available on children with DS and COVID-19. Some reports have highlighted a few cases of children who had one or more comorbidities, including cardiovascular anomalies, obesity, and/or obstructive sleep apnoea [7,8], but these reports were based on only 55 and 4 participants, respectively.

This study aims to describe the epidemiological and clinical characteristics of COVID-19 in children with DS, including risk factors for a severe course of illness, compared to those in the general paediatric population. In this study, severe COVID-19 is assessed holistically by comparing the rate of hospitalisation (including the rate of mechanical ventilation and intensive care unit (ICU) admission), acute respiratory distress syndrome (ARDS), and shortness of breath, in addition to mortality. Due to the limited data availability, control data (children with COVID-19 from the general population) were only available from the US. As our data on children with DS were drawn from countries with varying income levels, we explored the differential course of paediatric COVID-19 cases from low-to-middle-income countries (LMICs) and from high-income countries (HICs) [9].

## 2. Materials and Methods

### 2.1. T21RS DS Survey

We used data from the Trisomy 21 Research Society (T21RS) international survey, the largest survey of individuals with DS who had COVID-19. Caregivers and clinicians caring for people with DS provided information about the symptoms, outcomes, and existing health conditions of COVID-19 patients using an anonymous online format. The clinician surveys also asked about COVID-related treatments and medical complications. From all paediatric cases under 18 years of age reported to T21RS from 9 April 2020 and 22 October 2020, after excluding patients with missing relevant information such as age, gender, or clinical information, our sample included 328 paediatric COVID-19 patients with DS. We excluded duplicated participants identified based on age, gender, and country, and other specific demographics. Because this study relies on data from the first wave of the pandemic, 29% (95/328) of the paediatric cases reported in the T21RS dataset had not been tested for COVID-19 but were symptomatic (Table A1). For validation purposes, we further compared the data to those generated from participants who completed the T21RS survey during a later phase of the pandemic (22 October 2020, to 2 August 2021; 133 additional paediatric COVID-19 patients with DS). Figure A1 shows the distribution of when COVID-19 patients with DS were entered into the T21RS database.

The survey was implemented through REDCAP [10,11], a survey and database management system, and was hosted at Emory University. The survey was disseminated through clinical routes (e.g., Down Syndrome medical interest group listservs and health service providers), Down Syndrome associations in the US, India, Spain, UK, France, Italy, Germany, Brazil, and Spanish-speaking Latin America, and DS registries (e.g., NIH DS-Connect^®^), as well as via the T21RS website. Each participating institution obtained IRB/ethics approval (Institutional Review Board Statement). The study was performed according to the Declaration of Helsinki and national guidelines and regulations for data privacy, and all participants who completed the questionnaires provided informed consent. All data were anonymised according to good clinical practice guidelines and data protection regulations.

### 2.2. US COVID-NET Comparison Data

To contrast the COVID-19-related signs, symptoms, and medical complications with those observed in the general population, we compared the hospitalised T21RS DS cases to published results from paediatric cases reported by the COVID-19–Associated Hospitalisation Surveillance Network (COVID-NET). COVID-NET is a US population-based surveillance system that collects data on laboratory-confirmed COVID-19–associated hospitalisations [12]. From 1 March to 25 July 2020, 576 hospitalised paediatric COVID-19 cases were reported to the COVID-NET10.

### 2.3. Statistical Analyses

We used descriptive statistics to show the demographic information, outcomes, and COVID-19 symptoms and comorbidities of the participants included in our analyses. Categorical data were described as frequencies (percentages), and quantitative variables were provided as the mean (SD) or median (IQR). We used Fisher’s exact tests to compare characteristics of the COVID-NET controls to those observed in COVID-19 patients with DS. Due to limited data availability, we only had control data from the US (COVID-NET). To detect possible country-specific differences in the healthcare system and access to hospitals, we stratified the COVID-19 patients with DS from the T21RS survey by the income level of the country of residence. Countries were categorised as HICs (USA, Canada, Western Europe, Argentina, Australia) or LMICs (Brazil, India, Iran, Egypt, Belarus, South Africa, Mexico, Costa Rica, Bangladesh, Nepal) based on the World Bank income classification system [13].

To identify risk factors associated with adverse outcomes of COVID-19 in children with DS, we estimated associations between potential risk factors and admission to hospital, diagnosis of acute respiratory distress syndrome (ARDS), and the presence of shortness of breath using logistic regression analyses. All association analyses were restricted to symptomatic COVID-19 cases from the T21RS survey data, controlling for data source (caregiver versus clinician survey), living situation, level of intellectual disability, age, gender, and country of residence. 

All data analyses were conducted using R (version 4.0.0).

## 3. Results

This retrospective cohort study included 328 paediatric patients <18 years of age with DS and COVID-19. Of the individuals for whom we had relevant information, 38.8% (127/325) were hospitalised. The mean age was 9.55 (SD = 5.24) years (8.47 (SD = 4.99) for non-hospitalised and 11.37 (SD = 5.07) for hospitalised respondents. Of the patients for whom country information was available, 42.9% (137/319) were from India, 17.6% (56/319) were from Brazil, 11.3% (36/319) were from the UK, 9.4% (30/319) were from the USA, 3.8% (12/319) were from Spain, and 15% (48/319) were from other countries (Table 1). Note that some information was missing for certain patients; percentages reflect the portion of individuals for whom data were available. A description of the children’s ethnicities stratified by country of residence is presented in Table A2. The age distribution was slightly different between the T21RS individuals with Down Syndrome and the CDC controls without Down Syndrome (Table A3). The CDC controls involved more newborns <1 year of age (CDC controls: 27.3%, T21RS: 4.3%), and the proportion of 5–11 year-olds was higher among the study population with Down Syndrome (CDC controls: 16.8%, T21RS: 36.3%). The gender distribution was comparable between both studies (CDC controls: 50.7% male, T21RS: 47.4% male).

Among children with DS and COVID-19, hospitalised children had more respiratory signs and symptoms than those that were not hospitalised (Table A4). Among those hospitalised with COVID-19, children with DS had significantly higher rates of cough (53.7% vs. 29.5% in controls), fever (94.3% vs. 54% in controls), nasal signs/congestion (65.0% vs. 23.7% in controls), and shortness of breath (60.2% vs. 22.3% in controls) than the control group (Table 2). Hospitalised children with DS did not experience a significantly different prevalence of abdominal symptoms in comparison to the control group.

Medical complications were more prevalent among children with DS than among controls. Children with DS had significantly higher rates of pneumonia (36.9% vs. 11.1% controls), ARDS (32.8% DS vs. 1.9% controls), and acute renal failure (17.3% DS vs. 2.9% controls). There was no significant difference in the prevalence of Multisystem Inflammatory Syndrome between COVID-19 patients with and without DS.

Children with DS had similar rates of ICU admission (38.4% DS vs. 33.2% controls) but were more often put on mechanical ventilation (24.8% DS vs. 5.8% controls). Hospitalised patients with DS had a higher mortality rate compared with controls (5.0% DS vs. 0.5% controls), but all deaths occurred among children from LMIC.

Participants completing the T21RS survey during a later phase of the pandemic (22 October to 2 August 2021; *n* = 133) included fewer hospitalised cases (25 (20%)), but among those who were hospitalised, proportions of ICU admissions and medical complications were similar to individuals with DS from the earlier phases of the pandemic (Table A5 and Table A6). 

Next, we examined potential risk factors for severe COVID-19 (shortness of breath, hospitalisation, and ARDS) among COVID-19 patients with DS. Rates of hospitalisation by age group and country income category can be found in Figure 1. A description of the prevalence of comorbidities is provided in Table A7. Older age (OR 1.75 for 5 additional years [1.37–2.23]), obesity (OR 2.27 [1.12–4.59]) and epilepsy (OR 3.97 [1.65–9.58]) were all significant risk factors for hospitalisation among COVID-19 patients with DS (Figure 2 and Table A8); older age (OR 2.29 for 5 additional years [1.26–4.15]) and thyroid disorder (OR 3.40 [1.03–11.9]) were significant risk factors for ARDS; and older age (OR 1.31 for 5 additional years [1.05–1.63]) was a significant risk factor for shortness of breath.

The pandemic has posed challenges to healthcare systems globally, and the nature of the pandemic response may differ depending on the country income status. We stratified our analysis by country income level to explore this. First, we examined hospitalisation rates by age group and country income status. Patients with DS from LMICs had a higher rate of hospitalisation (48.5%) than the sample as a whole (39.0%), except for among children under 5 years of age. In LMICs, the hospitalisation rate increased with age group, consistent with our finding that older age was a risk factor for hospitalisation in our sample (Figure 2). In HICs, the hospitalisation rate was highest for children under 5, but was similar for other age groups. The age distribution of hospitalised COVID-19 patients with DS from HICs versus LMICs was similar during the later phases of the pandemic (Figure A2).

We compared the symptoms and signs of COVID-19 between the control group (all hospitalised) and hospitalised children with DS from the HICs and the LMICs. The prevalence of cough (82.6% in HICs, 46.5% in LMICs, 29.5% in controls), fever (91.3% in HICs, 95% in LMICs, 54% in controls), nasal congestion (56.5% in HICs, 66.7% in LMICs, 23.7% in controls), and shortness of breath (78.3% in HICs, 55.6% in LMICs, 22.4% in controls) was significantly higher among children with DS, independent of the country income status (although the latter was higher in HICs than in LMICs) (Table 2).

On average, children with DS spent more days in the hospital than children from the general population, independent of their country income status (controls: median of 2.5 days, HICs: median of 8 days, LMICs: median of 10 days). The prevalence of ICU admission was not different among children with DS from LMICs (43.4%) or HIC (17.4%) in comparison to controls (33.2%). The prevalence of mechanical ventilation was higher among children with DS from LMICs (27.2%) than among controls (5.8%) (*p* < 0.001), but for children with DS from HICs (14.3%), there was no statistically significant difference from controls. Lastly, with respect to mortality rates, all deaths among hospitalised children with DS were from LMICs, leading to estimates of mortality of 6.7% in LMICs, with no deaths occurring in either controls or children with DS from HICs. 

## 4. Discussion

Previous studies of individuals with DS suggest that, as in the general population, children do not face the same risk for COVID-19-related mortality as older adults. However, as severe COVID-19 cannot be assessed by mortality alone, we aimed to evaluate the rate of several indicators for severe COVID-19 (rate of hospitalisation, use of mechanical ventilation and ICU admittance, ARDS, shortness of breath, and mortality) among 328 children with DS and compare these to 224 children from the general population. 

### 4.1. Hospitalisation and Outcomes

In our study, hospitalised children with DS had significantly higher rates of cough, fever, nasal signs, and shortness of breath than the control group. This could be interpreted as a more severe COVID-19, as reflected by the following evidence of poor outcomes: (1) significantly higher rates of medical complications, including pneumonia, ARDS, and acute renal failure, and (2) need for mechanical ventilation (24.8% vs. 5.8% in controls). Mortality rates for children with DS were much lower than for adults with DS, as has also been reported in general for the paediatric population [12]. We could not examine risk factors for mortality because there were very few deaths among children with and without DS. Our results in children with DS are broadly in line with previous works on COVID-19 in adults with DS and with the literature, which suggests that people with DS may be more susceptible to viral and bacterial infections in terms of symptoms, complications, and a severe course of illness [4,5,7,14,15,16,17].

Risk factors for a severe course of COVID-19 are well-documented for the adult population with [2] and without DS [18,19], but less so for the paediatric population. In our sample, many of the documented risk factors for COVID-19 morbidity were found to be significant, including older age, epilepsy, and obesity. However, several known risk factors for a severe course of COVID-19 were not found to be statistically significant, including male sex [20] and chronic lung disease [2]. This may reflect a lack of statistical power and underscores the need for further study.

Pre-existing epilepsy as a risk factor for severe COVID-19 outcomes has not been established for those without DS [21,22]. However, there is a consensus that management of COVID-19 in people with epilepsy compared with those without may be more complicated, as would be true for those with DS and epilepsy. Some examples of complications include drug–drug interactions [23], the use of certain epilepsy medications that may affect the immune system [22], fever related to COVID-19 that may increase the risk of seizures and lead to increased hospitalisation [21], or increased need for health care or emergency care that may lead to higher exposure to the virus [21,22]. In those with and without DS, seizures could be a neurological consequence of COVID-19, as reviewed in Niazkar et al. [24]. However, further insights into the link between pre-existing epilepsy as a risk factor for COVID-19 outcomes in DS would benefit from further research. The same can be said of the risks associated with thyroid disorder—a common comorbidity of DS. It is possible that infection affected the thyroid function in children with DS, but the mechanism cannot be determined from the data used in this study. However, both SARS and COVID-19 are known to be associated with thyroid abnormalities in adult patients. SARS patients have been found to have decreased serum T3, T4, and TSH levels [25]. Thyroid hormone dysfunction affects the clinical course of COVID-19 as it increases mortality in critical illnesses such as ARDS, a major complication of COVID-19 [24]. Angiotensin-converting enzyme 2 (ACE2) requires the transmembrane serine protease 2 (TMPRSS2) to enter cells. In humans, the TMPRSS2 enzyme is encoded by the TMPRSS2 gene located on chromosome 21 (21q22.3 (OMIM 602060)). TMPRSS2 in turn forms a complex with the ACE2 receptor, enabling the virus to penetrate the cell surface directly and efficiently [26]. ACE2 and TMPRSS2 are highly expressed in the thyroid gland [26]. Further investigation is needed to elucidate whether the impact of SARS-CoV-2 on thyroid function is one of several factors in the development of ARDS in people, particularly children, with DS and/or primarily due to the overexpression of TMPRSS2 via a chromosome 21 gene-dose mechanism.

Finally, results varied by country income status for some attributes. The rate of hospitalisation for children younger than 5 years of age with DS in HICs, and for the US controls, was particularly high compared to LMICs. This may be a result of neonates being infected in hospitals immediately after birth [12]. Equally, the higher availability of resources could allow children under 5 years of age in HICs to be hospitalised for minor non-COVID-related symptoms or signs of illness in general, regardless of whether or not they have severe COVID-19. The high rate of hospitalisation in this age group in HICs (compared with children with DS from LMICs) might not, therefore, be indicative of a more severe course of illness.

### 4.2. Implications for Policy and Practice

Our results suggest that children with DS are more likely to develop severe COVID-19 than the general paediatric population. However, it is recognised that a child’s welfare also needs to be considered. Shielding children may protect them from being infected, but it will also impact their access to education and interaction with peers and could have an adverse impact on their development and wellbeing. While this is true for all children, it is particularly important for children with DS, who are likely to be affected even more by social isolation and the loss of their daily routine and therapies. These risks must therefore be balanced with those of protecting children from the infection itself. Parents, health professionals, and policy makers may therefore wish to consider the risks of infection to an individual child taking account of their likely exposure to the infection, existing comorbidities, and family circumstances.

While we found that the risk of severe COVID-19 increased with the presence of certain comorbidities for our population of interest, we also found that children with DS are, in general, at increased risk of severe outcomes from this disease. We therefore recommend, when vaccination is rolled out for paediatric populations, that children with DS be prioritised.

There are insufficient data currently for a conclusive understanding of the lifelong effects of COVID-19. Thus, a mild course of the disease may have long-term implications, and children with DS who experience a less severe course of illness or who are not hospitalised may still require annual reviews to ensure that their condition has not deteriorated.

### 4.3. Limitations and Areas for Future Research

One notable limitation of our study is that control data were only available from hospitalised [10] paediatric patients in the US, whereas the T21RS study samples came from many different countries; thus, country-specific differences could have biased our results. Some countries had more COVID-19 cases than others, health care systems differ substantially between countries, and some countries experienced a surge in cases later than others, meaning that they had more time to prepare and develop treatment strategies. Furthermore, the reason for admission to hospital and ICU could differ between countries. To address this potential bias, we conducted a comparison stratified by patients from LMICs and HICs. However, we are aware that there are also substantial differences between HICs, and we did not have a comparison group from LMICs. Furthermore, the age distribution of the children with and without DS differed and, as we had access only to the summary statistics, we could not adjust our association analyses for age. Subsequent international studies should explore a comparison with controls from a range of countries when suitable data become available. In addition, most of our cases from LMICs were from India (137/202), making the generalisation of findings to other LMICs difficult.

When reporting the ethnicity of patients, we used WHO categories, but the way in which ethnicity was reported varied among countries. We had limited data from Black, indigenous populations, Asian, and other children representing minority ethnic groups in HICs and may have overlooked risks unique to those groups.

Another limitation of our study is that most COVID-19 patients from the T21RS and the US COVID-NET data were reported during the first wave of the pandemic, when there were few treatment strategies available and mechanical ventilation was often the only treatment for severe COVID-19 cases with respiratory distress. However, a validation analysis that included COVID-19 patients with DS from 22 October 2020 to 2 August 2021 showed that hospitalised COVID-19 patients with DS from this later phase of the pandemic had similar outcomes and severity to those from earlier phases of the pandemic, strengthening our main conclusions. Another limitation of this study is that not all of the COVID-19 cases were confirmed by COVID-19-specific laboratory tests. Given that our survey was launched at the beginning of the pandemic when testing was scarce, we would have lost 29% of our cases (particularly from Spain, France, and Italy) if we were to include only individuals with laboratory-confirmed COVID-19. Furthermore, our previous publication on COVID-19 patients with DS indicated that there was no difference in the clinical presentation of those who were or were not tested for COVID-19 [2], providing an added level of confidence for the clinical diagnosis even when laboratory tests were not reported. In addition, as this study is based on an online survey that was completed by clinicians as well as family members/caregivers of the individuals with DS, we could not include any objective laboratory outcome measures. Furthermore, the T21RS survey data are a convenient sample of individuals with DS who had COVID-19 and not necessarily representative of all COVID-19 patients with DS. Therefore, the hospitalisation rates (39% of the children with DS were hospitalised) should be interpreted with caution and are likely affected by an overrepresentation of patients with severe COVID-19. More attention should also be paid to non-hospitalised and asymptomatic paediatric COVID-19 patients with DS, who could experience long-term effects of COVID-19. Thus, future research should aim to determine if the symptoms of non-hospitalised children with DS differ from those of the general paediatric population. Because our control data pertained to hospitalised patients only, this analysis lay beyond the scope of this investigation. Future research must also investigate the negative psychosocial and developmental impact of being home-bound during the COVID-19 pandemic on children with DS: emerging survey-based studies are already underway to document potential outcomes in terms of weight gain, regression, risk of osteoporosis from lack of exercise, and so forth. Our understanding of COVID-19 will be further enhanced by studying asymptomatic cases: as such, it will be important to include data from asymptomatic paediatric patients with COVID-19 (and their caregivers and clinicians) in future surveys now that testing has become more widely available.

## 5. Conclusions

Hospitalised children with DS had a higher prevalence of several symptoms of COVID-19 (cough, fever, nasal signs, shortness of breath) than hospitalised US controls; a higher prevalence of medical complications from COVID-19 (pneumonia, ARDS, acute renal failure), particularly among children with DS from LMICs; and were more likely to be placed on mechanical ventilation. Older age, obesity, and epilepsy were significant risk factors for hospitalisation; older age and thyroid disorder were significant risk factors for ARDS; and older age was a significant risk factor for shortness of breath. Efforts should be made to monitor the health of children and young people with DS during the ongoing pandemic and to report any COVID-19 signs and symptoms in a timely manner, especially for those who have comorbidities which are risk factors for severe COVID-19. In addition, our results emphasise the importance of vaccinating children with DS as soon as they become eligible.

## Figures and Tables

**Figure 1 jcm-10-05125-f001:**
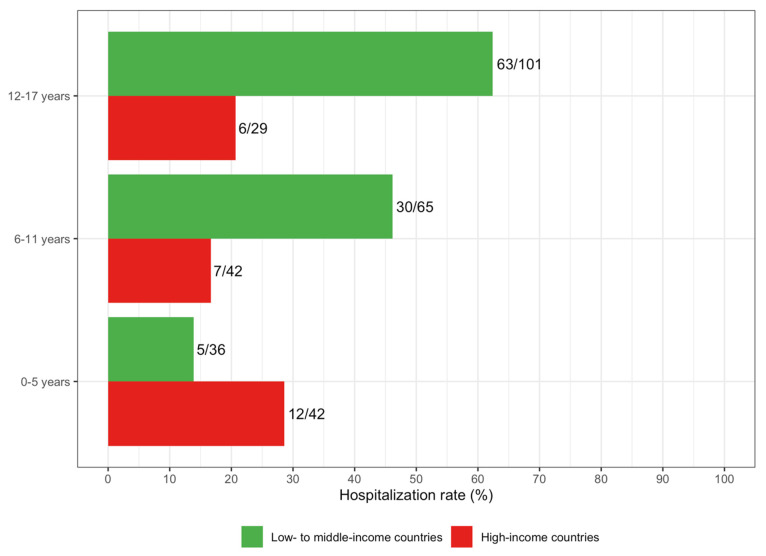
Hospitalisation rate of children with DS and COVID-19 in low to middle-income countries (Brazil, India, Iran, Egypt, Belarus, South Africa, Mexico, Costa Rica, Bangladesh, Nepal) and high-income countries (USA, Canada, Western Europe, Argentina, Australia).

**Figure 2 jcm-10-05125-f002:**
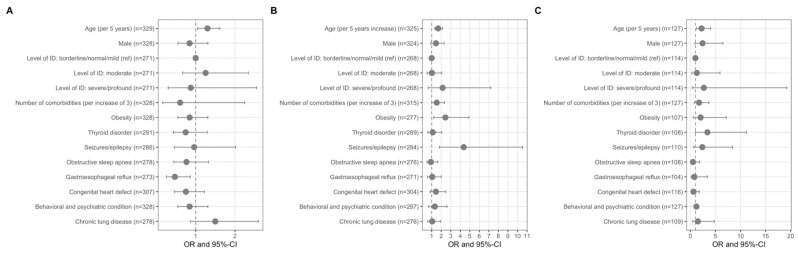
Risk factors associated with adverse outcomes of COVID-19 in children with DS. Associations with (**A**) shortness of breath, (**B**) hospitalisation, and (**C**) acute respiratory distress syndrome (ARDS) in symptomatic COVID-19 patients with DS from the T21RS survey estimated in adjusted logistic regression models (odds ratios (OR) and 95% confidence intervals (95% CI)). Associations with age and gender were adjusted for the data source (caregiver versus clinician survey) and associations with living situation, level of IDD, and comorbidities were adjusted for age, gender, data source, and country of residence. Abbreviations: IDD, intellectual and developmental disabilities; ref, reference category.

**Table 1 jcm-10-05125-t001:** **T21RS** study characteristics grouped by hospital admission.

	Overall	Not Hospitalised	Hospitalised
*n*	328	197	127
Additional information through follow-up (%) ^#^	29 (8.8)	25 (12.7)	4 (3.1)
Country (%)			
India	137 (42.9)	52 (27.1)	85 (69.1)
Brazil	56 (17.6)	44 (22.9)	11 (8.9)
United Kingdom	36 (11.3)	28 (14.6)	8 (6.5)
United States	30 (9.4)	22 (11.5)	7 (5.7)
Spain	12 (3.8)	9 (4.7)	3 (2.4)
other	48 (15.0)	37 (19.3)	9 (7.3)
Age (mean (SD))	9.55 (5.24)	8.47 (4.99)	11.37 (5.07)
Male (%)	173 (52.7)	97 (48.5)	74 (58.3)
Ethnicity (%)			
South Asian	142 (43.2)	54 (27.4)	88 (69.3)
White	110 (33.4)	89 (44.2)	21 (16.5)
Latin American	31 (9.4)	23 (11.7)	7 (5.5)
Black	6 (1.8)	3 (1.5)	3 (2.4)
Arab	1 (0.3)	1 (0.5)	0 (0.0)
West Asian	1 (0.3)	1 (0.5)	0 (0.0)
Admixed	12 (3.6)	9 (4.6)	3 (2.4)
Unknown	26 (7.9)	19 (9.6)	5 (3.9)
Living situation (%)			
Living at home with family	304 (98.1)	190 (97.9)	110 (98.2)
Living alone with support	1 (0.3)	0 (0.0)	1 (0.9)
Small group home with support	2 (0.6)	2 (1.0)	0 (0.0)
Residential care facility	2 (0.6)	1 (0.5)	1 (0.9)
Other	1 (0.3)	1 (0.5)	0 (0.0)
Type of trisomy 21 (%)			
Full/standard	274 (88.4)	167 (87.9)	105 (89.7)
Mosaic	29 (9.4)	17 (8.9)	11 (9.4)
Partial trisomy	2 (0.6)	2 (1.0)	0 (0.0)
Translocation	5 (1.6)	4 (2.1)	1 (0.9)
Level of intellectual disability (%)			
Borderline/normal/mild	83 (30.0)	59 (37.1)	23 (20.2)
Moderate	172 (62.1)	91 (57.2)	78 (68.4)
Severe/Profound	22 (7.9)	9 (5.7)	13 (11.4)
Admitted to hospital (%)	127 (39.2)	0 (0.0)	127 (100.0)
Days in hospital (mean (SD)) *	11.03 (5.18)	NA	11.03 (5.18)
Admitted to ICU (%) *	48 (38.4)	0 (0)	48 (38.4)
Days in ICU (mean (SD)) *	6.91 (2.52)	NA	6.91 (2.52)
Mechanical ventilation (%) *	28 (16.1)	0 (0.0)	28 (24.8)
Clinical situation at last evaluation (%)			
Currently in hospital with symptoms	38 (12.5)	0 (0.0)	37 (31.4)
Died	4 (1.3)	0 (0.0)	4 (3.4)
Not currently in hospital but with symptoms	45 (14.8)	34 (18.6)	10 (8.5)
Other	12 (3.9)	9 (4.9)	3 (2.5)
Recovered from COVID-19	197 (64.8)	132 (71.1)	64 (54.2)
Tested positive but still no symptoms	8 (2.6)	8 (4.4)	0 (0.0)

* Only answered if admitted to hospital; ^#^ Information that was missing in the original report was imputed by information provided through a follow-up survey (clinician follow-up for the family survey and family follow-up for the clinician survey, see methods for more details); percentages were calculated after excluding missing information. NA: not applicable.

**Table 2 jcm-10-05125-t002:** Severity of COVID-19 among hospitalised individuals with and without DS stratified by high-income and low to middle-income countries.

A. Signs and Symptoms Related to COVID-19											
	CDC controls		T21RS Individuals with Down Syndrome			T21RS Individuals with Down Syndrome, High Income Countries			T21RS Individuals with Down Syndrome, Low-to-Mid Income Countries		
		*n*		*n*	*p* Value (Comparison with Controls)		*n*	*p* Value (Comparison with Controls)		*n*	*p* Value (Comparison with Controls)
Cough, *n* (%)	66 (29.5)	224	66 (53.7)	123	0.005	19 (82.6)	23	<0.001	47 (47.0)	100	0.003
Fever, *n* (%)	121 (54.0)	224	116 (94.3)	123	0.001	21 (91.3)	23	<0.001	95 (95.0)	100	<0.001
Nasal congestion ^1^, *n* (%)	53 (23.7)	224	80 (65.0)	123	<0.001	13 (56.5)	23	0.002	67 (67.0)	100	<0.001
Shortness of breath, *n* (%)	50 (22.3)	224	74 (60.2)	123	<0.001	18 (78.3)	23	<0.001	56 (56.0)	100	<0.001
Abdominal pain, *n* (%)	42 (18.8)	224	23 (18.7)	123	1	3 (13.0)	23	0.776	20 (20.0)	100	0.879
Vomiting/nausea, *n* (%)	69 (30.8)	224	23 (18.7)	123	0.063	3 (13.0)	23	0.092	20 (20.0)	100	0.059
Diarrhoea, *n* (%)	27 (12.1)	224	18 (14.6)	123	0.621	6 (26.1)	23	0.098	12 (12.0)	100	1
B. Other Indicators for a Severe Course of COVID-19											
		*n*		*n*	*p* Value (Comparison with Controls)		*n*	*p* Value (Comparison with Controls)		*n*	*p* Value (Comparison with Controls)
Hospitalisation length of stay, median (IQR)	2.5 (4)	208	10 (7)	119	n/a	8 (9)	22	n/a	10 (7)	97	n/a
ICU admission, *n* (%)	69 (33.2)	208	47 (38.5)	122	0.506	4 (17.4)	23	0.041	43 (43.4)	99	0.099
Mechanical ventilation, *n* (%)	12 (5.8)	207	28 (24.8)	113	<0.001	3 (14.3)	21	0.149	25 (27.2)	92	<0.001
Death, *n* (%)	1 (0.5)	208	4 (5.0)	80	0.025	0 (0.0)	20	1	4 (6.7)	60	<0.001
C. Prevalence of Medical Complications among Hospitalised Patients ^2^											
		*n*		*n*	*p* Value (Comparison with Controls)		*n*	*p* Value (Comparison with Controls)		*n*	*p* Value (Comparison with Controls)
Pneumonia, *n* (%)	23 (11.1)	208	24 (36.9)	65	<0.001	5 (71.4)	7	<0.001	19 (32.8)	58	<0.001
Acute Respiratory Syndrome, *n* (%)	4 (1.9)	208	20 (32.8)	61	<0.001	1 (33.3)	3	0.070	19 (32.8)	58	<0.001
Acute renal injury/acute renal failure, *n* (%)	6 (2.9)	208	18 (17.3)	104	<0.001	2 (10.0)	20	0.149	16 (19.0)	84	<0.001
Multisystem Inflammatory Syndrome, Kawasaki-like disease, *n* (%)	9 (10.8)	83	1 (2.2)	46	0.164	1 (33.3)	3	0.361	0 (0.0)	43	0.057

^1^ T21RS survey: this sign refers to “nasal signs” rather than nasal congestion. ^2^ T21RS survey: medical complications were only recorded in surveys answered by clinicians.

## Data Availability

The data used in this study can be obtained from the websites of the Trisomy 21 Research Society (T21RS) and the US Centres for Disease Control (US-CDC).

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
