# Peer review of "COVID-19 in Children with Down Syndrome: Data from the Trisomy 21 Research Society Survey"

_jcm, 2021, doi:10.3390/jcm10215125_

Round 1

Reviewer 1 Report

This comprehensive international study on COVID19 in children with Down syndrome provided much needed information on the clinical presentation and the risk factors of COVID 19 in this special population.  The manuscript is well-organized, clear and understandable. Conclusions provide critical data for rational guidelines, but are balanced with caveats and explanation of limitations. This study was only possible with the cooperation of many investigators in different disciplines, and is an example of important research that can be accomplished in a short period of time with clinical usefulness for a vulnerable population.

I have a few minor questions and edits:

  1. Under Materials and Methods, T21 DS Survey, line 2   " … individuals with DS who had COVID-192 ." ( delete 2)
  2. Likewise, Discussion 4.1 Hospitalization and Outcomes , line 6  "...much lower than for adults with DS2, …" (delete 2)
  3.  Discussion 4.1 Hospitalization and Outcomes, paragraph 2, " Risk factors for a severe course of COFID-19 are well-documented for the adult population, …" Suggest … "adult GENERAL population" (for clarity).
  4.  Discussion 4.1 Hospitalization and Outcomes, paragraph 3,  "Pre-existing epilepsy as a risk factor for infection..." This point was unclear. Either provide elaboration or eliminate reference to infection,
  5. Section 4.1, paragraph 3, line 19: "ACE" should be spelled out if this is the first appearance of this term.
  6. On the same  line : …"(TMPRSS2) TMPRSS2..." (delete the duplicate)
  7. Section 4.3, paragraph 2 lines2 & 3: " We had limited data concerning about..."  Use one word or the  other not both.

Author Response

Please find responses to all reviewer comments in the attached letter 

Reviewer 2 Report

Please see attachement.

Author Response

(The authors gave the same response as above.)
